# Element and radionuclide concentrations in soils and wildlife from forests in north-east England with a focus on species representative of the ICRP's Reference Animals and Plants

Catherine L. Barnett[1], Nicholas A. Beresford[1,2], Michael D. Wood[2], Maria Izquierdo[3], Lee A. Walker[1],
Ross Fawkes[2]

[1]UK Centre for Ecology & Hydrology, Lancaster Environment Centre, Library Avenue, Bailrigg, Lancaster, Lancashire, LA1 4AP, UK
[2]School of Environment & Life Sciences, Peel Building, University of Salford, Manchester, M5 4WT, UK
[3]University of Nottingham, Sutton Bonington Campus, Sutton Bonington, Leicestershire, LE12 5RD, UK; current address: Institute of Environmental Assessment and Water Research, 18–26 Jordi Girona, 08034 Barcelona, Spain

*Correspondence to*: Catherine L. Barnett (clb@ceh.ac.uk)

**Abstract.** There are international recommendations that the environment (i.e. wildlife) is assessed for the potential impact of releases of ionising radiation. The transfer of radionuclides to wildlife from media (e.g. soil, water) is usually described using the whole-organism concentration ratio ($CR_{wo-media}$) and a number of assessment models use these values to estimate radiation exposure and risk to wildlife; however, there are many gaps in knowledge. This paper describes a study conducted in 2015-2016 to sample terrestrial wildlife, soil and water from two forests in north-east England. Sampling was targeted towards species representative of the International Commission on Radiological Protection's (ICRP) terrestrial Reference Animals and Plants (RAPs): Wild Grass (Poaceae family), Pine Tree (Pinaceae family), Earthworm (Lumbricidae family), Bee (Apidae family), Rat (Muridae family), Deer (Cervidae family) and Frog (Ranidae family); opportunistic sampling of plant and fungi species was also conducted. The dataset comprises stable element concentrations for 30 elements, radionuclide activity concentrations for K-40 and Cs-137, and radionuclide and stable element concentration ratios. These data have significantly increased the number of $CR_{wo-media}$ values available for the ICRP RAPs and will contribute to the development of the databases underpinning the ICRP's environmental protection framework. Data will be included in the international database of wildlife transfer parameters for radioecological models and hence are likely to contribute to model developments in the future.

All data and supporting documentation are freely available from the Environmental Information Data Centre (EIDC; https://eidc.ac.uk/) under the terms and conditions of the Open Government Licence (Barnett et al., 2020 https://doi.org/10.5285/8f85c188-a915-46ac-966a-95fcb1491be6).

## 1 Introduction

Over the last twenty years there has been a step change in international radiological protection, requiring assessment of environmental (i.e. wildlife) impacts of ionising radiation releases rather than assuming that protecting humans ensures protection of other species (ICRP 2007; IAEA 2014a). Consequently, various assessment models have been developed to estimate radiation exposure and risk to wildlife (e.g. Beresford et al. 2008a; Brown et al. 2016). These models require an approach to estimate radionuclide activity concentration in organisms if this is unknown (Beresford et al. 2008b). The transfer of radionuclides to wildlife is usually described using the whole-organism concentration ratio ($CR_{wo-media}$), where, in the case of terrestrial organisms, $CR_{wo-soil}$ is estimated as:

$$CR_{wo-soil} = \frac{\text{Activity concentration in whole-organism (Bq kg}^{-1}\text{ fresh mass)}}{\text{Activity concentration in soil (Bq kg}^{-1}\text{ dry mass)}} \qquad [1]$$


Compilations of $CR_{wo\text{-}media}$ values have been published by the International Atomic Energy Agency (IAEA 2014b) and the International Commission on Radiological Protection (ICRP 2009). These data compilations were facilitated by the creation of an on-line database of $CR_{wo\text{-}media}$ values (Copplestone et al. 2013; http://www.wildlifetransferdatabase.org/), which has continued to be updated (Brown et al. 2016). The available models and data compilations typically use a system

of simplified (often called 'reference') organisms (e.g. transfer data maybe collated for generic mammals, fish, trees etc.). Even so there are many organism-radionuclide combinations requiring $CR_{wo\text{-}media}$ values and large gaps in our knowledge. For instance, there were only data available for about 50% of the 1500 $CR_{wo\text{-}media}$ values required to populate the most recent version of the ERICA Tool (Brown et al. 2016), data are especially sparse for the ICRPs suggested Reference Animals and Plant (RAPs; ICRP 2008) (ICRP 2009) and some elements (IAEA 2014b)). As one approach to fill data

gaps the ICRP (ICRP 2009) suggested selecting a series of sites from which samples of as many RAPs as possible could be collected; this has subsequently been referred to as the 'Reference Site' concept (Thørring et al. 2016). In Barnett et al. (2014) we presented the first such study (together with the accompanying dataset Barnett et al. 2013), which was conducted in north-west England; subsequently studies have been published for sites in Spain, Norway and the Ukrainian Chernobyl Exclusion Zone (Guillén et al. 2017, 2018; Thørring et al. 2016; Beresford et al. 2018a, 2020). In addition to

radionuclides, these studies presented data for stable element concentrations; stable elements are increasingly being used as proxies for radionuclides in parameterising models (Beresford 2010; IAEA 2014b; Copplestone et al. 2013).

This paper describes a study conducted in 2015-2016 to sample terrestrial wildlife and associated soil samples from two forest sites in north-east England (Fig. 1) sampled using a similar protocol to that described in Barnett et al. (2014) and

subsequently adopted by Guillén et al. (2018) and Beresford et al. (2020) for samplings in Spain and Ukraine. Samples were analysed to determine concentrations of stable elements (Ag Al, As, Ba, Be, Ca, Cd, Co, Cr, Cs, Cu, Fe, I, K, Li, Mg, Mn, Mo, Na, Ni, P, Pb, Rb, Se, Sr, Ti, Tl, U, V and Zn) the majority of which have radioisotopes which need to be considered in radiological assessments, and the gamma-emitting radionuclides $^{40}K$ and $^{137}Cs$. Whole-organism concentration ratios ($CR_{wo\text{-}media}$) have been estimated from these data. The complete dataset associated with this study is

available from https://doi.org/10.5285/8f85c188-a915-46ac-966a-95fcb1491be6 (Barnett et al. 2020).

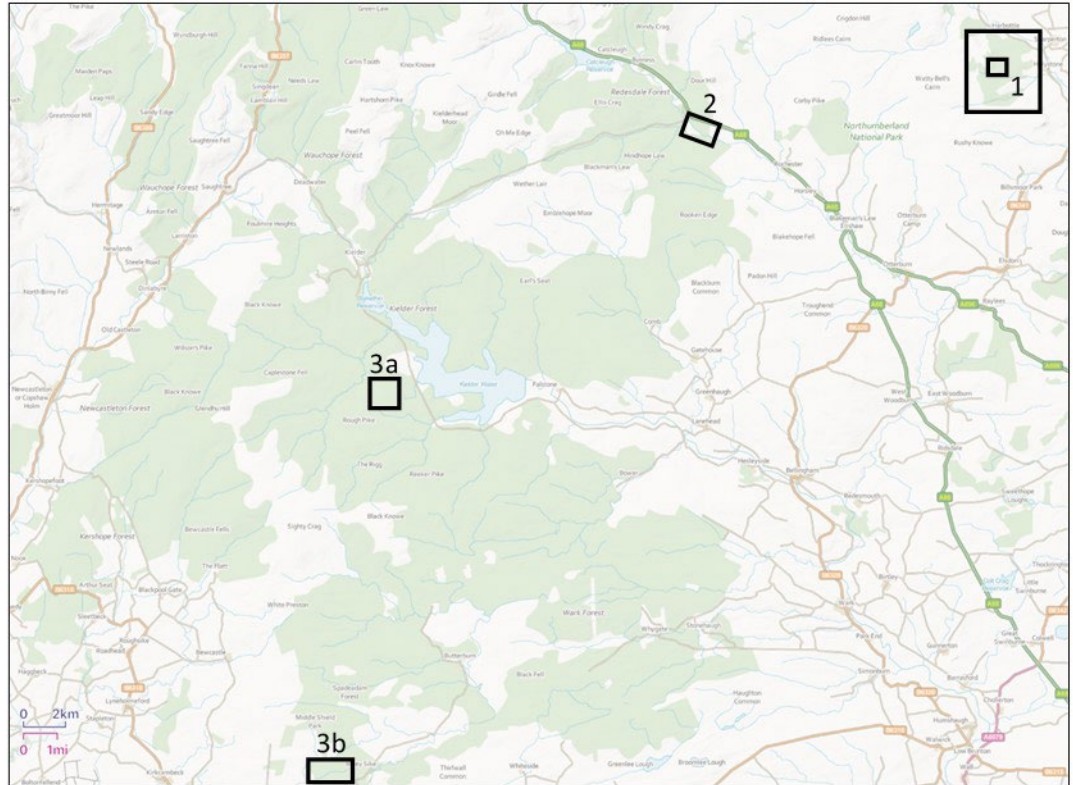

Fig. 1. Location of sampling sites. Site 1 (inner square) identifies the Holystone main sampling area, Site 1 (outer square) identifies the Holystone wider sampling area (from which Roe deer, fungal fruiting body, berry and some vegetation samples were collected). Site 2 identifies the Kielder main sampling area. Sites 3a and 3b identify the Kielder deer sampling areas (and those of the associated fungal fruiting body and some vegetation samples); note site 3b is referred to as Spadeadam in Barnett et al. (2020). Contains Ordnance Survey data © Crown copyright and database right 2020.

## 2 Materials and methods

### 2.1  Site descriptions

An overview of the sampling sites, 'Holystone Woods' and 'Kielder Forest' located in north-east England (Fig. 1), are given below.

### 2.1.1 Holystone Woods

Holystone Woods (Ordinance Survey National Grid Reference 3940 6030, total area *circa* 8.7 km$^2$, altitude *circa* 130 m to 360 m) is located within the Northumberland National Park; it borders the Ministry of Defence Otterburn training area (Forestry Commission England 2016) (Site 1 on Fig. 1). Three Sites of Special Scientific Interest (SSSIs) surround the area (Holystone Burn Wood, Holystone North Wood and Harbottle Moors). It is a coniferous forest plantation mainly established between 1954-1974 (Forestry Commission England, 2016); the dominant tree species are *Pinus sylvestris* and *Pinus contorta,* with some *Picea* spp. and *Larix* spp., and some other coniferous and broadleaf species (e.g. *Betula* spp.) present. Soil types vary and are a complex mixture dominated by peaty surface water gleys and podzolic ironpans. There is an abundance of *Pteridium* spp. in open places, and the understory is dominated by *Deschampsia flexuosa* and *Vaccinium myrtillus* with some *Calluna vulgaris* and *Sphagnum* species.

Our main sampling area (*circa* 0.06 km$^2$ in area) at Holystone Woods was within a mixed coniferous plantation, which also contained a few small broadleaf trees with an understory of grasses (predominantly *Molinia caerulea*) and some

sedge species and shrubs (e.g. *Vaccinium* spp.). Additional samples were also collected from the surrounding forest (marked on Fig. 1 as 'Holystone wider sampling area').

### 2.1.2 Kielder forest

Kielder Forest is also located in Northumberland; it is the largest man-made woodland in England (*c*. 650 km$^2$). Soils within Kielder vary considerably, with peat often being a major constituent (Forestry Commission England 2016). Our

main sampling area (*circa* 0.16 km$^2$ in size; Site 2 on Fig. 1) was situated in in the northern section of Kielder in an area referred to as Redesdale Forest (Ordinance Survey National Grid Reference (NGR) 3760 6010; 50 km$^2$, altitude 60-550m) (Forestry Commission 1951). Redesdale Forest, and Kielder Forest as a whole, is dominated by *Picea sitchensis* with a few other coniferous species, including *Larix* spp., and some broadleaf species (e.g. *Betula* spp.); the majority of coniferous trees were planted between 1930 and 1960 (Forestry Commission England 2011). The understory is mainly

*Molinia caerulea* and *Calluna vulgaris*. As for Holystone, additional samples were also collected from outside of the main sampling area; these sites are identified on Fig. 1 as 3a and 3b (site 3b is referred to as Spadeadam (the name of the forest 'block') in Barnett et al. (2020).

### 2.2 Sample collection and preparation

Sampling was targeted towards species representative of the ICRPs terrestrial RAPs: Wild Grass (Poaceae family), Pine

Tree (Pinaceae family), Earthworm (Lumbricidae family), Bee (Apidae family), Rat (Muridae family), Deer (Cervidae family) and Frog (Ranidae family). It was not possible to sample species representative of the Duck (Anatidae family). By-catch of *Myodes glareolus* and *Bufo bufo* were also retained for analyses. Additional plant and fungi species were collected when it was thought that these may contribute to the diet of the sampled deer.

The species analysed together with their relationship to both the relevant ICRP RAP (ICRP 2009) and also IAEA broad wildlife group (IAEA 2014b) are listed in Table 1. Appendix Table A1 summarises the elements and radionuclides determined in each sample type; the accompanying dataset (Barnett et al. 2020) provides details of all samples (including masses and dimensions where appropriate) including those collected and retained but not analysed.

**Table 1.** Species sampled and analysed during this study.

| Latin name | Common name | IAEA Broad Wildlife Group | ICRP RAP | Number of individuals analysed[*] |
|---|---|---|---|---|
| *Bombus hortorum* | Garden bumblebee | Arthropod | Bee | 7[#] |
| *Bombus pascuorum* | Common carder bee | Arthropod | Bee | 4[#] |
| *Bombus terrestris* | Buff-tailed bumblebee | Arthropod | Bee | 63[#] |
| *Capreolus capreolus* | Roe deer | Mammal | Deer | 6 |
| *Aporrectodea caliginosa* | Grey worm | Annelid | Earthworm | 5[#] |
| *Dendrodrilus rubidus* | Red wiggler | Annelid | Earthworm | 10[#] |
| *Lumbricus rubellus* | Red earthworm | Annelid | Earthworm | 5[#] |
| *Rana temporaria* | European common frog | Amphibian | Frog | 6 |
| *Rana sp.* [spawn] | Frog sp. [spawn] | Amphibian | Frog | 3 |
| *Bufo bufo* | European toad | Amphibian | n/a | 6 |
| *Apodemus sylvaticus* | European wood mouse | Mammal | Rat | 12 |
| *Myodes glareolus* | Bank vole | Mammal | n/a | 12 |
| *Abies alba* | Silver fir | Tree | Pine Tree | 3 |
| *Picea sitchensis* | Sitka spruce | Tree | Pine Tree | 4 |

| | | | | |
|---|---|---|---|---|
| *Pinus sylvestris* | Scots pine | Tree | Pine Tree | 3 |
| *Agrostis giganteana* | Black bent | Grasses and Herbs | Wild Grass | 4 |
| *Deschampsia flexuosa* | Wavy hair-grass | Grasses and Herbs | Wild Grass | 6 |
| *Molinia caerulea* | Purple moor grass | Grasses and Herbs | Wild Grass | 6 |
| *Alopecurus myosuroides* | Black grass | Grasses and Herbs | n/a | 2 |
| *Achillia millefolium* | Common yarrow | Grasses and Herbs | n/a | 1 |
| *Calluna vulgaris* | Heather | Shrub | n/a | 12 |
| *Centaurea nigra* | Black knapweed | Grasses and Herbs | n/a | 3 |
| *Cirsium vulgare* | Spear thistle | Grasses and Herbs | n/a | 3 |
| *Cirsium palustre* | Marsh thistle | Grasses and Herbs | n/a | 3 |
| *Digitalis purpurea* | Common foxglove | Grasses and Herbs | n/a | 9 |
| *Erica tetralix* | Cross-leaved heath | Shrub | n/a | 4 |
| *Juncus effuses* | Common rush | Grasses and Herbs | n/a | 9 |
| *Luzula luzuloides* | White wood rush | Grasses and Herbs | n/a | 1 |
| *Polytrichum commune* | Common hair moss | Lichens and Bryophytes | n/a | 1 |
| *Vaccinium myrtillus* | Billberry | Shrub | n/a | 1 |
| *Vaccinium sp.* | Billberry sp. | Shrub | n/a | 2 |
| Fungal fruiting bodies[^] | n/a | Fungi | n/a | 22[#] |

*For these species further samples were collected but were not analysed, information (e.g. mass and dimensions) is available from Barnett et al. 2020); [#]some samples were analysed as composite samples; [^]see Sect. 2.2.13 for a list of species, some species were analysed as composite samples, n/a information not applicable; soil (n=47) and water (n=3) samples were also collected.

At Holystone Woods the majority of samples were collected from the 0.06 km² study area with the exception of the three *Capreolus capreolus* which were collected at distances of less than 2 km to the north and east of the main study site within the 'Holystone wider sampling area' on Fig. 1. Samples of vegetation species and also fungal fruit bodies which may be ingested by deer (see Sect. 4) were collected from the wider sampling area as identified in Fig. 1. As for Holystone, the majority of samples from Kielder Forest were collected from the main sampling area. Three *C. capreolus* were collected

from elsewhere within Kielder Forest (see Sites 3a and 3b on Fig. 1). Samples of species which may contribute to the diet of deer (see Sect. 4) were also collected from these areas.

Details of the sampling methods for each sample type are given in Sect. 2.2.1- 2.2.13; locations (NGRs) of all individual sampling sites were recorded using a hand held GPS (accuracy approximately 5 m).

**2.2.1 Soil**

Sixteen soil samples (approximately 15 cm x 15 cm x 10 cm deep (as specified in ICRP 2009)) were collected in August 2015 from both the main Kielder and main Holystone sampling sites; locations corresponded to the sampling sites of the animals and plants collected. A further 15 soils were collected in January 2016 from areas where *C. capreolus* were sampled from within *c.* 0.5 km² of where the deer were shot (0.5 km² is the typical home range of a female *C. capreolus*

(Barnett et al. 2014)); these comprised, three samples from Site 3a, three from Site 3b and nine from the wider Holystone sampling area. Although collected by spade the soil samples were trimmed by knife such that a consistent profile was obtained. Prior to processing surface vegetation and detritus were removed, with the root mat (where present) being retained in the sample.

Sub-samples were removed (ensuring each was representative of the complete soil profile) from each fresh soil sample for determination of pH, loss on ignition (LOI) and dry matter content. The remaining sample was then divided into two sub-samples which were air-dried at 20ºC to avoid loss of volatile elements. Once dry, one sub-sample was ground in an agate mill, prior to ICP-MS analysis; the second sample was ground in a Christie and Norris mill or soil sieve (depending upon organic matter content) to approximately 2 mm for subsequent gamma analysis.


The methods described by Allen (1989) were used to determine the pH (in distilled water) and LOI (ashing at 450ºC for four hours) of all 47 soil samples collected.

### 2.2.2 Water

Three water samples (each of approximately 1 litre) were collected from seasonal ponds in the Kielder main sampling site on 16 March 2015. The samples were collected from three areas within the amphibian trapping area (Sect. 2.2.6); *Rana* sp. spawn samples were also collected at the same time. All water samples were stored at $4^0$C following collection; once in the laboratory all samples were filtered through Whatman 541 filters. A 50 mL aliquot of each of the filtered water samples was adjusted to 1% $HNO_3$ (Baker ultrex II) prior to ICP-MS analyses; the remaining filtered samples underwent gamma analyses.

### 2.2.3 *Bombus* spp. (Bee spp.)

Bees belonging to the *Bombus* genus were collected from both main sampling sites between 17 and 21 August 2015 using pan traps (21 cm diameter plastic bowls) which were coloured white, fluorescent yellow or fluorescent blue (see Westphal et al. (2008) for a description of the trapping method). At both the Holystone and Kielder sites traps were placed on the ground surface close to small mammal trapping points (see Sect. 2.2.8). The pan traps were filled with de-ionised water to which a small amount of non-perfumed washing-up liquid was added (to prevent evaporation) and covered with a wire cage (a plasticised hanging plant basket) to prevent e.g. amphibian access; pan traps were emptied every day over the five day sampling period. After sieving, individual bees were separated from other invertebrates into pre-weighed scintillation vials and stored frozen (-20ºC). Species from each site were identified by an expert (Carvell pers. comm.) (see Table 1). The fresh masses (FM) of a sub-set of individual bees, from each species, were recorded and the samples then freeze-dried with subsequent dry mass (DM) being recorded. Dried samples were homogenised in an agate mortar to avoid contamination by metals. The majority of the bees were acid digested and analysed by ICP-MS as bulked samples of individuals of the same species from a given site. The only exceptions were a large *B. terrestris* queen and a single *B. hortorum* collected from Holystone that were analysed as individuals.

### 2.2.4 *Capreolus capreolus* (Roe deer)

Three female adult *C. capreolus* were obtained via the culling scheme operated by the Forestry Commission from within the wider Holystone sampling area (see Fig. 1) on 3 November 2015; one male animal was obtained from Site 3a within Kielder Forest and two animals (one male and one female) were obtained on 5 November 2017 from Site 3b.

The live-mass of all the deer was recorded by the Forestry Commission upon sampling. All animals were collected from site promptly and dissected at UKCEH. The following organs and tissues were removed: liver, kidney, spleen, lung, thyroid gland, a hind-leg (dissected to obtain separate muscle and bone samples) and the gastrointestinal tract and its contents. The carcass from one animal (identified as Deer 6 in the accompanying dataset) was also divided down the spine and all muscle, fat and bone was removed from one-half of the carcass; the data obtained was used to estimate the total body mass of each of these components. Bone samples were cleaned of any residual soft tissue by placing in a beetle (*Dermestes maculatus*) colony.

All samples had their FM recorded and, where appropriate, were diced into approximately 1 cm pieces. They were retained frozen (-20ºC) prior to freeze-drying and subsequent analysis. The freeze-dried kidney, liver, gonad, muscle and bone

samples were homogenised and analysed by ICP-MS. Gamma analyses were performed on fresh rumen contents and fresh muscle samples.

### 2.2.5 Lumbricidae (Earthworms)

Earthworms, belonging to the Lumbricidae family, were collected at both the Kielder and Holystone main sampling sites from various locations between 20 and 21 August 2015 by digging to approximately 30 cm. After rinsing the earthworms in de-ionised water they were placed in aerated containers with damp tissue paper to allow for gut evacuation for 48 hours. Individuals from each site were identified by species, and adults, sub-adults and juveniles were separated. Composite samples were then created, two from Holystone and one from Kielder, each of these contained 5-10 adult individuals from the same species. The dimensions, FM and DM (after freeze-drying) for the individuals were recorded prior to bulking. Freeze-dried composite samples were homogenised in an agate mortar prior to analyses by ICP-MS.

### 2.2.6 *Rana temporaria* (European common frog) and *Rana* sp. spawn

Within the Kielder main sampling area, an amphibian trapping site (approx. 400 $m^2$) was erected in a 'boggy' area. A 1 m tall drift fence was constructed using plastic sheeting and wooden poles. The drift fence provides a physical barrier to amphibian movement within an area and amphibians then tend to follow the fence line. On each sampling visit, the full length of the drift fence was searched by four people, with the aim of hand capturing any amphibians present; however, searching by torchlight along the forestry tracks during darkness was more successful. In total four frogs were collected between 13 and 18 August 2015 from the Kielder main sampling area and three from Holystone (one of which was not analysed). After humane dispatch the animals were stored frozen (-20ºC) until further preparation.

The frogs had their FM recorded and then their gastrointestinal tract removed. The following organs and tissues were removed from each animal: gonads, liver, fat bodies, 'tissue containing the thyroid gland' where this was possible (this approach was taken due to the small size of the thyroid, see Guillén et al. 2018), a hind-leg (dissected to obtain both muscle and bone samples) and a composite sample comprising kidney, spleen and lung. Each carcass and hind-leg bone were placed in a beetle (*D. maculatus*) colony to clean the bone of all soft tissue. All the tissue samples had their FM recorded, were diced into small pieces (as appropriate) and retained frozen (-20ºC) prior to freeze-drying (DM was recorded in most cases) prior to ICP-MS analysis.

Three spawn samples were collected on 16 March 2015 at the Kielder main sampling site from the area of boggy ground where the amphibian trapping sheet was subsequently located. Each spawn sample was washed in de-ionised water and the sample FM was recorded. Each sample was then freeze-dried and the DM recorded. Following freeze-drying, samples were homogenised in an agate mortar prior to ICP-MS and gamma analyses.

### 2.2.7 *Bufo bufo* (European toad)

Sixteen toads were collected as described above for frogs on the 13 August (n=12) and 19 August (n=4) 2015. All except one (which was collected from the Holystone main sampling area) were collected from the Kielder main sampling area; they were all stored frozen (-20ºC) upon capture and a whole-body FM was recorded. Subsequently, six toads were randomly selected and prepared for analysis. Their gastrointestinal tract (together with its contents) was removed and the FM of the carcass (less gastrointestinal tract) was recorded. The carcass was then washed in de-ionised water; freeze-dried (DM recorded) and ashed in a muffle furnace for approximately 24 hours (using a stepped heating programme up to 400ºC). The six ash samples had their ash mass (AM) recorded and were subsequently analysed by ICP-MS. Toads

(and voles, see below) were ashed as they were by-catch and were outside the scope of the core study; ashing gave a simple sample preparation technique as there was no need to analyse individual tissues.

### 2.2.8 *Apodemus sylvaticus* (Wood mouse)

At both the Holystone and Kielder main sampling sites, 60 'Longworth' traps were placed along wall lines, beside fallen trees or along obvious small mammal tracks; they were placed in these positions as mice prefer to move along linear features (Spellerberg and Gaywood, 1993). At Holystone, the traps were placed at up to four separate locations within the main sampling area and at Kielder at six locations within the main sampling area. Depending upon the location traps were placed either in blocks of 12 or in short runs containing 4-12 traps.

Traps were baited overnight with oats, carrot and insect pupae; grass hay was added as bedding material. At both sites baited traps were left open for three days before trapping began to familiarise the animals to them. Once trapping began all traps were inspected each morning and then closed, they were re-baited and reset early each evening. Trapping at the Holystone main sampling site took place between 17 and 21 August 2015. At Kielder, trapping was only conducted at the main sampling site on 17 August 2015; subsequent trapping was unnecessary due to the high number of animals caught. Trapping was conducted in accordance with Natural England general licence WML-GL01 (Natural England 2017). Mice (and voles, Sect. 2.2.9) were euthanised immediately upon being found in a trap using the appropriate humane method given in Schedule 1 of the Animals (Scientific Procedures) Act (1986) (UK Parliament 1994). Any shrews (*Sorex* sp.) trapped were released.

A total of nine male and five female mice were trapped at Holystone and eight male and ten female mice at Kielder (two additional mice were caught but their sex was not recorded). Whole-body FM (including gastrointestinal tract and contents) and dimensions were recorded for all mice. Randomly selected mice (n=12, six from Holystone and six from Kielder) had their pelt and gastrointestinal tract (including contents) removed and the FM of the remaining carcass recorded; this was then washed in de-ionised water. The liver, gonads, a sample of 'tissue containing the thyroid gland' (see Guillén et al. 2018), a hind-leg (dissected for separate muscle and bone samples) and a composite sample made up of the spleen, kidney and lung were removed. Individual tissue FM was recorded and the samples were then stored frozen (-20$^{\circ}$C) prior to freeze-drying after which their DM was recorded. The remaining carcass (and hind-leg bone samples) was placed in a beetle (*D. maculatus*) colony to clean bone of remaining soft tissue; once clean the FM of the remaining bone from both the carcass and hind-leg was recorded and samples stored frozen (-20$^{\circ}$C) prior to freeze-drying after which their DM was recorded. The liver, 'tissue containing thyroid', muscle and bone samples were homogenised. The tissue samples from the 12 randomly selected mice were analysed by ICP-MS.

### 2.2.9 *Myodes glareolus* (Bank vole)

Voles were by-catch from the mice trapping activities. Thirteen voles were captured from the Holystone main sampling site and fourteen from the Kielder main sampling site; they were stored frozen (-20$^{\circ}$C) after being euthanised as described in Sect. 2.2.8. All voles had their whole-body FM recorded; the pelt and gastrointestinal tract (including contents) was then removed and the FM of the remaining carcass was recorded for each vole; they were then washed in de-ionised water. Of the 27 animals captured, six (three male and three female) from each main sampling site were randomly selected for ICP-MS analysis. These twelve carcasses were then freeze-dried (DM recorded) and ashed in a muffle furnace for approximately 24 hours, using a stepped heating programme up to 400$^{\circ}$C; the AM was then recorded prior to analysis by ICP-MS.

**2.2.10 *Abies alba*, *Pinus sylvestris* and *Picea sitchensis* (Silver fir, Scots pine and Sitka spruce)**

Species selected for sampling were those that were dominant at each site. Samples of heartwood core (taken using an increment borer), small branches (approximately 1 cm in diameter) and needles were collected from pine trees located at both the Kielder (three *P. sitchensis*); and Holystone (two *P. sylvestris* and one *A. alba*) main sampling areas on 20th August 2015 and from one tree (*P. sitchensis*) at Site 3b on 21st January 2016. Needles (and cones (if present)) were removed from the collected branches, the branches were then cut into small pieces (*circa* 1 cm length). The FM was recorded for all sampled tree components, which were then air-dried (approximately 20°C) prior to DM being determined. Branch, needle and cone samples were then finely ground using an agate mill or mortar (dependent upon size) and the heartwood core samples were manually size reduced to approximately 5 mm in diameter. ICP-MS analysis was conducted on all the needle and heartwood core samples; all the branch samples and two cone samples also underwent gamma analysis.

**2.2.11 *Molina caerulea*, *Agrostis giganteana* and *Deschampsia flexuosa* (Purple moor grass, Black bent and Wavy hair-grass)**

A total of sixteen samples were collected from Kielder (n=4 *A. giganteana*, n=3 *D. flexuosa* and n=3 *M. caerulea*) and Holystone (n=3 *D. flexuosa* and n=3 *M. caerulea*) main sampling areas on 20 August 2015; both leaf and flower stems were collected to within *c.* 1 cm of the soil surface. Species selected were those that were abundant at each site. All the samples were stored cool (4°C) prior to transport to the laboratory where their FM was recorded; they were then air-dried (approximately 20°C) and subsequently their DM was recorded. All samples were finely ground using an agate mill or mortar (dependent upon sample size) and analysed by ICP-MS. Fifteen samples having a DM >5 g also underwent gamma analysis.

**2.2.12 Herbaceous plants and shrubs**

Numerous opportunistic vegetation samples were collected from both the Kielder and Holystone main sampling areas with most samples being collected on 20 August 2015 (Table 1); species selected for sampling were abundant at each site. Some species were also sampled from where *C. capreolus* were obtained (see Sect. 2.2.4); these samples were collected either on 22 January 2016 or 24 October 2016.

All samples were stored cool prior to transport to the UKCEH; upon receipt, species collected were positively identified and had their FM recorded. The samples were then air-dried (approximately 20°C) and once dry, the DM was recorded. Subsequently, they were all finely ground using an agate mill or mortar (dependent upon size) and analysed by ICP-MS; if, following ICP-MS analysis, there was remaining sample >5g (DM) gamma analysis was also conducted.

**2.2.13 Fungal fruiting bodies**

Opportunistic samples of fungal fruiting bodies were collected, placed in paper bags (to avoid sample degradation) and stored cool (4°C) prior to transport to the UKCEH where species were positively identified. Generally single bulked samples of different species were collected from the sites:

Holystone wider sampling area (24 October 2016) - *Galerina vittiformis*, *Clitocybe* sp., *Collybia* sp., *Hygrophoropsis aurantiaca*, *Hygrophorus* sp., *Laccaria laccata*, *Russula emetica*, *Suillus* sp.;

Kielder Site 3a (24 October 2016) - *Galerina sp., Collybia sp., Cortinarius sp., H. aurantiaca, Mycena epipterygia, Psathyrella sp.*;

Kielder Site 3b (23 October 2016) - *Calvatia excipuliformis*, *Galerina sp.* (n=2), *Gymnopilus sp.*, *H. aurantiaca*, *Hygrophorus sp.* (n=2), *L. laccata*.

After recording the FM of the individual samples they were oven-dried at approximately 60ºC and their DM recorded. Individual samples were then finely ground to approximately 2 mm using either an agate mortar or electric coffee grinder depending upon their size. With the exception of the single *C. excipuliformis* all fungal fruiting bodies were bulked into six replicates by sampling location and, where possible, by feeding strategy. If, after bulking, the sample size was too small for gamma analysis the sample volume was increased by mixing with cornflour (an inert material of a similar mass). Gamma analysis was then conducted on all samples.

### 2.3  Sample digestion

Acid digestions were undertaken at the University of Nottingham to determine elemental concentrations in soils, plant material and animal tissues. Alkaline extractions with tetramethylammonium hydroxide (TMAH) were also undertaken to determine concentrations of iodine in soils, plants and (where sample size was sufficient) animal tissues.

### 2.3.1 Soil

Acid digestion was undertaken by weighing approximately 0.2 g of dried ground soil into a Savillex™ vial, adding concentrated Primar grade $HNO_3$ (4 mL) and heating at 80°C overnight using a teflon-coated graphite hot block; this first step was undertaken due to the comparatively high organic matter contents in a number of soils. Then concentrated Primar grade HF (2.5 mL), $HNO_3$ (2 mL) and $HClO_4$ (1 mL) were added. A stepped heating programme up to 160°C overnight was applied to fully digest silicate and oxide phases. The dry residue was re-constituted after warming with 2.5 mL ultrapure MilliQ water and 2.5 mL $HNO_3$ and the final volume made up to 50 mL. The Standard Reference Material NIST SRM 2711a Montana soil in duplicate and five blanks were all digested in a similar manner to check the accuracy and precision of the digestion and analysis methods. All the digests were diluted 1-in-5 before analysis.

To determine iodine concentrations, an alkaline extraction was undertaken using TMAH. A portion of soil (approximately 1 g) was weighed into polypropylene tubes and 10 mL of 10% TMAH were added. The soil suspensions were heated at 90°C for 24 h, and then centrifuged at 3500 rpm for 30 min. A 1 mL aliquot was pipetted off and diluted 10-fold to give a final TMAH concentration of 1% for further analysis. Four blanks were prepared in a similar manner.

### 2.3.2 Plant material

Microwave assisted acid digestion was conducted on a sub-sample of dried-ground plant material (approximately 0.2 g), which was weighed into digestion vessels and 6 mL concentrated Primar grade $HNO_3$ added. The samples were digested using a Multiwave PRO Anton Paar microwave reaction system, with heating at 140°C for 20 minutes and further cooling to 55°C for 15 minutes. Once the digestion was complete, the samples were made to a final volume of 20 mL. Full dissolution was achieved in all cases. Digestion of five replicates of Standard Reference Material NIST 1573a Tomato Leaves and seven blanks were all undertaken in a similar manner to check accuracy and precision of the digestion and analysis methods. The elemental recoveries for the certified reference material were typically >90%. Prior to analysis, the acid digests were diluted 1-in-15 to give a final matrix of 2% $HNO_3$.

For microwave assisted TMAH extraction, a portion of plant material (approximately 0.2 g) was weighed into digestion vessels and 5 mL of 5% TMAH was added. The samples were microwave digested, heating at 110°C for 30 minutes followed by a cooling step at 40°C for 12 minutes. The extracts were transferred to polypropylene centrifuge tubes and

made up to a final volume of 25 mL to give a final TMAH concentration of 1%. Before analysis, the samples were centrifuged (3500 rpm for 30 min) and a 5 mL aliquot of the supernatant transferred into ICP auto-sampler tubes for analysis. Four replicates of CRM NIST 1573a Tomato Leaves and four blanks were all prepared in a similar manner to check the efficiency of the extraction and total iodine recovery.

TMAH extraction is not able to fully digest plant material and variable degrees of dissolution were observed. However, the analysis of CRM NIST 1573a showed a recovery of 78-85% iodine, calculated on the basis of a non-certified iodine concentration of 0.85 mg kg$^{-1}$. This suggests that, even though plant tissues were not fully dissolved, the TMAH extraction solubilises a sufficient proportion of iodine provided that the material is finely ground; samples with coarse particle size i.e. pine tree cores showed poor dissolution rates and therefore the reported iodine concentrations may be underestimates.

**2.3.3 Animal tissues**

For microwave assisted acid digestion, where available a portion of dried-ground animal tissue (approximately 0.2 g) was weighed into digestion vessels and a mixture of 3 mL Primar grade $HNO_3$ + 3 mL MilliQ ultrapure water + 2 mL 30% v/v $H_2O_2$ was added. The samples were allowed to froth for 30 minutes in uncovered vessels before microwave digestion at 140°C for 20 minutes. Once the digestion was complete, the extracts were made to a final volume of 20 mL. Seven

replicates of CRM NIST 1577c bovine liver and 14 blanks were all prepared in a similar manner. Prior to analysis, the acid digests were diluted 8-fold to give a final $HNO_3$ concentration of approx. 2%. Recoveries of typically 90-110% were obtained for the certified elements for NIST 1577c. Full dissolution was achieved for all untreated samples but variable dissolution rates were observed for ashed samples subjected (e.g. vole and toad carcasses) where the presence of undissolved black particles in the solutions suggested that the digestion was incomplete.


For TMAH extraction of animal tissues a portion of dried-ground tissue (approximately 0.2 g) was weighed into polypropylene tubes and 5 mL of 5% TMAH was added. The tubes were then placed in an oven (90°C±3°C for 5 h), occasionally tapping and swirling to help dissolution. The extracts were allowed to cool, diluted with ultrapure water to a final volume of 25 mL and then centrifuged at 3500 rpm for 25 minutes. An aliquot of 5 mL of supernatant was then

transferred to ICP tubes ready for analysis. Exceptions were the thyroid gland from the deer, for which a further 200-fold dilution was required to bring concentrations within the calibration range (0.5-100 µg L$^{-1}$ iodine). The majority of samples were fully or largely digested; however, a number of them showed undissolved white particles, most likely undigested bone. In addition, as reported above for acid digests, ashed samples showed poor dissolution rates. For the three frogs for which a 'tissue containing thyroid' was available, there was insufficient sample for other tissues types to conduct I

analyses.

**2.4 Stable element analysis**

With the exception of the water samples, which were analysed at UKCEH, multi- element analysis of soils, plants and animal tissues was conducted at the University of Nottingham using an iCAP-Q ICP-MS (Thermo Fisher Scientific, Bremen, Germany).


For acid digests, the instrument was run employing three operational modes including (i) a collision-cell (Q cell) using He with kinetic energy discrimination (He-cell) to remove polyatomic interferences, (ii) standard mode (STD) in which the collision cell is evacuated and (iii) hydrogen mode (H$_2$-cell) in which H$_2$ gas is used as the cell gas. Samples were introduced from an autosampler (Cetac ASX-520) incorporating an ASXpress™ rapid uptake module through a PEEK

nebulizer (Burgener Mira Mist). Internal standards were introduced to the sample stream on a separate line via the ASXpress unit and included Ge (10 µg L$^{-1}$), Rh (10 µg L$^{-1}$) and Ir (5 µg L$^{-1}$) in 2% trace analysis grade (Fisher Scientific, UK) HNO$_3$. External multi-element calibration standards (Claritas-PPT grade CLMS-2 from SPEX Certiprep Inc., Metuchen, NJ, USA) included Ag, Al, As, Ba, Be, Ca, Cd, Co, Cr, Cs, Cu, Fe, K, Li, Mg, Mn, Mo, Na, Ni, P, Pb, Rb, S, Se, Sr, Tl, U, V and Zn, in the range 0 – 100 µg L$^{-1}$ (0, 20, 40, 100 µg L$^{-1}$). A bespoke external multi-element calibration
solution (PlasmaCAL, SCP Science, France) was used to create Ca, Mg, Na and K standards in the range 0-30 mg L$^{-1}$. Phosphorus, B and S calibration utilised in-house standard solutions (KH$_2$PO$_4$, K$_2$SO$_4$ and H$_3$BO$_3$). In-sample switching was used to measure B and P in STD mode, Se in H$_2$-cell mode and all other elements in He-cell mode. Peak dwell times were 10 mS for most elements with 150 scans per sample. Sample processing was undertaken using Qtegra™ software (Thermo-Fisher Scientific) utilizing external cross-calibration between pulse-counting and analogue detector modes when
required.

Iodine analysis in the alkaline extracts was undertaken separately using a 1% TMAH matrix for both standards and samples. The concentrations were determined in standard (STD) mode (evacuated collision cell) using Re (5 µg L$^{-1}$) in 2% trace analysis grade as internal standard to correct for suppression or enhancement effects. The instrument was
calibrated (0-100 µg L$^{-1}$ $^{127}$I) using synthetic chemical solutions diluted from NaIO$_3$ stock solution.

Where the amount of sample provided was >0.3 g, both extractions were conducted; in those cases where the amount of available sample was limited, acidic digestion was prioritised unless the samples were 'tissue containing thyroid' (Guillén et al. 2018). For each element, extraction form and sample type (i.e. soil, plant or animal tissue), limits of detection
(LODs) were calculated as three times the standard deviation of the reagent blanks. The suite of elements reported for soil, vegetation and animal samples is: Ag, Al, As, Ba, Be, Ca, Cd, Co, Cr, Cs, Cu, Fe, I, K, Li, Mg, Mn, Mo, Na, Ni, P, Pb, Rb, Se, Sr, Ti, Tl, U, V and Zn.

Water analysis was conducted at UKCEH for the full suite of trace metals (As, Be, Ce, Cr, Cu, Li, Ni, Pr, Sb, Sn, Ti, V,
Zn, Ba, Cd, Co, Cs, La, Mo, Pb, Rb, Se, Sr, U and W) on a Perkin Elmer ICP-MS. Analysis of a core suite of elements (Al, Ca, K, Mn, Si, Fe, Mg, Na and B) was also conducted by ICP-OES (Perkin Elmer DV 7300). Results for quality control and duplicate samples analysed alongside test samples were within expected ranges.

**2.5 Gamma analysis**

To determine the activity concentration of $^{40}$K and $^{137}$Cs, soil, water, fungal fruiting body and selected plant (see Sect.
2.2.10, 2.2.11, 2.2.12) and animal (see Sect. 2.2.4, 2.2.6) samples were analysed in suitably sized containers (*circa* 130 or 700 mL (Dormex Ltd.) or 15 mL petri dishes) on hyper-pure germanium detectors using an efficiency calibration suited to sample density. Dried-ground soil samples were analysed for two days; dried-ground plant samples and fungal fruit body samples were analysed for four days and animal samples were analysed as fresh material (with the exception of frogspawn, which was analysed dry) for up to four days. Size of the available sample determined if gamma-analyses were
conducted, the minimum sample size was 15 mL.

The detectors were calibrated for efficiency against standards of various density and volume using a certified reference solution (National Physics Laboratory, R08-04 'Mixed nuclide standard solution'; http://www.npl.co.uk/upload/pdf/mixed_nuclide_standard_solutions.pdf), which covers an energy range of
approximately 59-1850 keV. The resultant spectra were analysed using the Canberra Apex software and the estimated

activity concentrations (and a two sigma counting error) were decay corrected to the day of sampling. Replicate analyses were conducted on random samples and process blanks for quality assurance purposes and the laboratory regularly participates in IAEA (https://nucleus.iaea.org/sites/ReferenceMaterials/Pages/Interlaboratory-Studies.aspx ), IARMA (http://www.iarma.co.uk/services/analytical-proficiency-tests-2/) and US Department of Energy MAPEP (https://mapep.inl.gov/) proficiency testing schemes.

## 2.6 Calculation of whole-organism concentrations for mammals and amphibians

To calculate whole-organism concentrations for Roe deer, small mammals and amphibians we used the total masses and elemental concentrations determined in the analysed tissues. The approach and assumptions used are similar to those used in previous papers (Barnett et al. 2014, Guillén et al. 2018).

Roe deer whole-organism stable element concentrations were calculated assuming bone, muscle, kidney, liver and thyroid comprised the whole-organism. To calculate total concentrations in a given tissue the measured total organ mass, or, in the case of bone and muscle, estimated total tissue masses were used. Total bone and muscle masses were estimated using the average percentage contributions of these tissues to the whole-body mass as determined from the animal on which the half carcass dissection was performed in this study and similar results from three Roe deer from an earlier study (see Barnett et al. 2014). The exception was for the deer subjected to the half carcass dissection in this study, which used estimates of total muscle and bone mass derived for this animal. For $^{137}$Cs it has been assumed that activity concentrations in muscle are representative of those in the whole-organism (Beresford et al. 2008c; Yankovich et al. 2010).

Wood mice whole-organism stable element concentrations have been calculated using an estimate of concentrations in total bone and of total muscle and concentrations in the liver (and for iodine concentrations in 'tissue containing thyroid'). For European common frogs, whole-organism stable element concentrations were calculated in the same manner as for Wood mice with the exception that gonad masses and concentrations were also used. In Barnett et al. (2014) we demonstrated that the contribution of gonads to the whole-organism concentrations of elements in Wood mice was minimal and hence we did not analyse mice gonads in this study. It was not possible to calculate whole-organism concentrations of iodine for the frogs due to the lack of results for tissues other than the sample containing the thyroid. For some animals an element may not have been detectable in all analysed tissues. If this was the case the LOD value was used for that element-tissue. If the contribution of an element to the total body content was estimated to be ≥10% from tissues with concentrations below the LOD, then the whole-organism concentration is reported as a 'less than' value in Barnett et al 2020. If the contribution from tissues with concentrations below the LOD was <10% of the total body content, then the estimated whole-organism concentration was assumed to be a reasonable approximation.

For Bank voles and Common toads no data manipulation was necessary as whole (ashed) carcasses were analysed.

## 2.7 Calculation of whole-organism concentration ratios

Whole-organism concentration ratios were calculated using equation [1] with soil concentrations being the arithmetic mean value for the appropriate sampling site (e.g. for all samples types obtained from the main sampling areas the mean soil concentration over the sampling area was used); soils used to calculate $CR_{wo-soil}$ values are identified within the accompanying dataset (Barnett et.al. 2020). For mammals and amphibians whole-organism concentrations (as estimated by the manner described above) were used to calculate whole-organism concentration ratios. For earthworms and bees whole-organisms were analysed to determine their elemental concentrations. For plants and fungi $CR_{wo-soil}$ values were

estimated for the components analysed; for the RAP Pine tree the ICRP geometry is the trunk (i.e. heartwood, ICRP 2008) although $CR_{wo-soil}$ values are also presented here for needles. For frogspawn, $CR_{wo}$ values were estimated as $CR_{wo-water}$ values. All $CR_{wo}$ values are expressed on a FM basis.

## 3. Dataset

All data associated with this study are available (as .csv files) from Barnett et al. (2020); the supporting documentation associated with the dataset contains descriptions of the contents of each .csv file which forms the dataset. In summary the data available from Barnett et al. (2020) are:

***Bombus spp.* (Bee spp.):**
- Stable element concentrations (mg kg$^{-1}$ (FM and DM)) for composite (single species) samples and for a sample
470        containing a single *B. hortorum* queen;
- Whole-organism concentration ratios (FM);
- FM and DM of bees sampled but not analysed.

***C. capreolus* (Roe deer):**
- Stable element concentrations (mg kg$^{-1}$ (FM and DM)) for hind-leg bone, muscle, liver, kidney, thyroid gland
475        and rumen contents;
- Stable element concentrations presented as 'mg per tissue' (i.e. the total amount (mg) of an element in a given tissue) for all tissue types other than rumen contents;
- Whole-organism stable element concentrations (FM);
- Whole-organism concentration ratios (FM);
- Radionuclide activity concentrations (Bq kg$^{-1}$ (FM and DM)) for $^{137}$Cs for muscle;
- Caesium-137 concentration ratios (calculated using the $^{137}$Cs activity concentrations (FM) measured in muscle).
- FM and DM of various organs and tissues sampled but not analysed.

**Lumbricidae (Earthworms):**
- Stable element concentrations (mg kg$^{-1}$ (FM and DM)) for composite (single species) samples;
- Whole-organism concentration ratios (FM);
- FM, DM and dimensions of earthworms sampled but not analysed.

***R. temporaria* (European common frog) and *Rana* sp. spawn:**
- Stable element concentrations (mg kg$^{-1}$ (FM and DM)) for hind-leg bone, muscle, liver, gonad and 'tissue containing thyroid' (see Guillén et al. 2018);
- Stable element concentrations presented as 'mg per tissue' for all the tissue types listed above other than 'tissue containing thyroid';
- Whole-organism stable element concentrations (FM);
- Whole-organism concentration ratios (FM);
- FM and DM of some organs and tissues (e.g. lung, spleen, renal organ and fat bodies etc.) sampled but not
495        analysed;
- Stable element concentrations (mg kg$^{-1}$ (FM and DM)) for *Rana* sp. spawn;
- Stable element concentration ratios (FM) for *Rana* sp. spawn (calculated using the stable element concentrations in water);
- Radionuclide activity concentrations (Bq kg$^{-1}$ (FM and DM)) for $^{40}$K and $^{137}$Cs in *Rana* sp. spawn.

*B. bufo* **(European toads)**:

- Whole-organism stable element concentrations (mg kg$^{-1}$ (FM and AM));
- Whole-organism stable element concentration ratios (FM);
- FM and DM of some organs and tissues (e.g. lung, spleen, renal organ, fat bodies and gastrointestinal tract (and contents) etc.) sampled but not analysed.

*A. sylvaticus* **(Wood mice):**

- Stable element concentrations (mg kg$^{-1}$ (FM and DM)) for hind-leg bone, muscle, liver and 'tissue containing thyroid' (see Guillén et al. 2018), no data for Be is presented as too few tissues had concentrations in excess of the LOD ($<1\times10^{-3}$ mg kg$^{-1}$ FM);
- Stable element concentrations presented as 'mg per tissue' for all tissue types listed above;
- Whole-organism stable element concentrations (FM);
- Whole-organism concentration ratios (FM);
- The FM and DM of some additional organs/tissues (e.g. kidney, spleen, lung, gonad, pelt) that were sampled but not analysed.

*M. glareolus* **(Bank voles):**

- Whole-organism stable element concentrations (mg kg$^{-1}$ (FM and AM));
- Whole-organism concentration ratios (FM);
- FM of voles sampled but not analysed.

*A. alba*, *P. sylvestris* and *P. sitchensis* **(Silver fir, Scots pine, Sitka spruce):**

- Stable element concentrations (mg kg$^{-1}$ (FM and DM)) for needles and heartwood (core);
- Radionuclide activity concentrations (Bq kg$^{-1}$ (FM and DM)) for $^{137}$Cs and $^{40}$K for the needles and branches;
- Stable element concentration ratios (FM);
- Caesium-137 concentration ratios (FM and DM);
- FM and DM for samples not analysed.

*M. caerulea*, *A. giganteana* and *D. flexuosa*:

- Stable element concentrations (mg kg$^{-1}$ (FM and DM));
- Radionuclide activity concentrations (Bq kg$^{-1}$ (FM and DM)) for $^{137}$Cs and $^{40}$K for samples large enough for analysis;
- Stable element concentration ratios (FM);
- Radionuclide concentration ratios (FM and DM) for $^{137}$Cs.

**Herbaceous species and shrubs** (see Table 1)**:**

- Stable element concentrations (mg kg$^{-1}$ (FM and DM);
- Radionuclide activity concentrations (Bq kg$^{-1}$ (FM and DM)) for $^{137}$Cs and $^{40}$K for samples large enough to analyse;
- Stable element concentration ratios (FM);
- Radionuclide concentration ratios (FM and DM) for $^{137}$Cs.

**Fungal fruiting bodies** (see Sect. 2.2.13 for a list of species sampled):

- Radionuclide activity concentrations (Bq kg$^{-1}$ FM and DM) for $^{137}$Cs and $^{40}$K for composite samples and also for a single sample of *C. excipuliformis*;
- Radionuclide concentration ratios for $^{137}$Cs (FM and DM).

**Soil**:

- Stable element concentrations (mg kg$^{-1}$ (DM));
- Radionuclide activity concentrations (Bq kg$^{-1}$ (DM)) for $^{40}$K and $^{137}$Cs;
- pH and LOI measurements for some samples.

**Water**:

- Stable element concentrations (mg kg$^{-1}$ (filtered))
- Radionuclide activity concentrations (Bq kg$^{-1}$ (filtered)) for $^{40}$K and $^{137}$Cs.

## 4. Results and discussion

This study has significantly increased the data available for the ICRP RAPs. For instance, ICRP (2009), the compilation of CR$_{\text{wo-media}}$ values for the ICRP RAPs, has no data for bees or frogspawn and data for only four elements for deer and

only five elements for frogs. The most numerous data in ICRP (2009) were for earthworms and, even for this RAP, data were only available for 17 elements compared to the approximately 40 elements considered in some radiological environmental assessment tools (e.g. Brown et al. 2016).

Detailed analyses and discussion of the data are outside the scope of this paper and consequently we provide an overview

only. As noted, the published ICRP compilation of CR$_{\text{wo-media}}$ values (ICRP 2009) had rather limited data making meaningful comparisons difficult. Where comparisons are possible then the data reported here and presented in ICRP (2009) are generally within the same range. There are some instances where the CR$_{\text{wo-media}}$ values reported here are at the lower end of the ICRP range[1], these include: Co and Sr for Reference Rat; Cs, Pb and Sr for Reference Wild grass and Pb for Reference Pine tree. Since the publication of ICRP (2009) a number of studies have been conducted using a protocol

similar to that described here in an attempt to provide data for the developing ICRP environmental protection framework. These studies were conducted in north-west England (Barnett et al. 2014), Mediterranean Spain (Guillén et al. 2018), the Ukrainian Chernobyl Exclusion Zone (Beresford et al. 2020) and Norway (Thørring et al. 2016). As, in some instances, these new data represent a larger dataset than considered in ICRP (2009) we have compared data from them to the data reported here for selected elements of radiological interest (Ag, Co, Cs, Fe, Pb, Se, Sr and U). In most instances, the data

reported here were within the range of those from the other studies. The exception was for Cs and Roe deer, where the CR$_{\text{wo-soil}}$ values for both stable Cs and $^{137}$Cs were about an order of magnitude higher for the sites reported here than those for the other four sites.

This study and those reported using a similar protocol (Barnett et al. 2014; Thørring et al. 2016; Guillén et al. 2018;

Beresford et al. 2020) have significantly increased the available data for the ICRP RAPs for a large number of elements. However, as the studies have tended to focus on stable elements and gamma-emitting radionuclides, some radionuclides which may need to be included in radiological environmental assessments have been neglected (e.g. only Thørring et al. (2016) considers $^{226}$Ra whilst Beresford et al. (2020) was the only study to include Am and Pu radioisotopes). In future studies using protocols similar to that described here should, resources allowing, consider a broader range of analyses to

encompass a wider range of elements or radiological concern.

Transfer parameters for radioecological models for both wildlife and human foodstuffs are increasingly being derived from measurements of stable elements using ICP-MS analyses (e.g. Copplestone et al. 2013; IAEA 2010, 2014b) as we

---

[1] Note ICRP (2009) does not present ranges – however those for the data reviewed by ICRP can be found here: http://www.wildlifetransferdatabase.org/downloadsummary.asp ('ICRP RAP tables (Feb 2011)')

have done in the study reported here. This has the advantage that multiple elements can be determined in a single analysis and for many elements it is the only feasible approach to obtaining data. These data are used in radioecological models assuming that stable element transfer parameters will be reflective of those for radioisotopes at equilibrium. However, we note that where $CR_{wo-soil}$ values from this study could be compared for stable Cs and $^{137}$Cs, that the values for $^{137}$Cs were consistently higher than those for stable Cs. Comparison was possible for 30 samples (Roe deer, grassy and herbaceous vegetation species, moss, heather species and pine trees), the ratio of $^{137}$Cs (*c.* 30 y physical half-life) to stable Cs $CR_{wo-soil}$ values ranged from 1.3 to 10.5 across these samples (median = 4.7). Similar observations for Cs, and to a lesser extent Sr (which also has a *c.* 30 y physical half-life), have recently been made at other sites (Barnett et al. 2014; Beresford et al. 2019, 2020; Thørring et al. 2016). The increasing number of similar observations questions the use of stable element $CR_{wo-soil}$ values within radiological assessments; radiological assessments typically aim to be conservative. However, as already noted, for many elements there is no other realistic approach to obtaining data on environmental transfer other than to use stable element data.

## 4.1 Use of data

The data described here will be added to the 'Wildlife Transfer Database' (Copplestone et al. 2013; http://www.wildlifetransferdatabase.org/) which supports ICRP and IAEA activities and provides a resource for the updating of models (e.g. Brown et al. 2016; Beresford et al. 2018b).

In total in this paper we report on the derivation of 3889 $CR_{wo-media}$ values for 30 elements (and 90 $CR_{wo-media}$ values for Cs-137) across a range of plant and animal species collected in temperate forest ecosystems. The study focussed on obtaining data for species falling within the definition of the ICRP's RAPs (ICRP 2008) given the lack of data for these (ICRP 2009). As noted above, this study makes a considerable contribution to addressing this lack of data and as such the results are, and will continue to be, used by the ICRP in the on-going development of their environmental assessment framework (e.g. see https://www.icrp.org/icrp_group.asp?id=92). Data are described here for elements for which $CR_{wo-media}$ values were previously lacking, for instance ICRP (2009) contains no data for iodine (nor does the more comprehensive compilation of $CR_{wo-media}$ values presented in IAEA (2014b)).

In wildlife assessment models, the concentrations of the vast majority of elements for terrestrial species are estimated via $CR_{wo-soil}$ values (e.g. Beresford et al. 2010). However, for logical reasons (Galeriu et al. 2003) activity concentrations of $^{3}$H and $^{14}$C are related to air not soil. Because of a lack of any data for P-isotopes when models were originally developed an assumption was made that $^{14}$C $CR_{wo-air}$ concentrations could be used as proxies (Copplestone et al. 2001; Beresford et al. 2008c; Brown et al. 2016). This assumption has never been tested and is likely not very robust (e.g. if terrestrial ecosystems are contaminated via discharges to sewers or rivers). The data reported here present some of the first $CR_{wo-soil}$ values for P, which will allow us to start to establish a more robust modelling approach for radioisotopes of this element. To our knowledge, the $CR_{wo-water}$ values we report for frogspawn are amongst the first available along with data we recently published for samples obtained from within the Chernobyl Exclusion Zone (Beresford et al. 2020).

Often data are available for radionuclides in wild animal tissues which are used in the human food chain (e.g. RIFE 2019). To be able to use such data in the derivation of transfer parameters for application in wildlife assessment models, a methodology to convert tissue specific concentrations to whole-organism concentrations is required. Yankovich et al. (2010) published tissue to whole-organism conversion factors and these were used to help populate the Wildlife Transfer Database (Copplestone et al. 2013). However, these factors are often based on few data. For instance, for terrestrial

mammals, the Ag, Cs, Se and U recommended conversion factors are each based on one value only and no values are presented for Co and Sr; no recommended conversion factors for amphibians are based on more than seven observations. The data discussed in this paper will contribute to improving the available conversion factors for both mammals and amphibians, including for elements for which Yankovich et al. present no data. Our data also enable a comparison of the transfer of radionuclides to the whole-organisms and the gonads of amphibians. Assessment of dose to gonads is of interest

as reproductive effects are a key endpoint when considering population level effects of exposure to ionising radiation (ICRP 2009); data for mammals were previously available (ICRP 2008; Barnett et al. 2014). Our data suggest little difference in the likely transfer of radionuclides to the gonad of amphibians compared to that of the whole-organism. Where there were differences (Ca, Mn, Pb, Sr and Ti) then gonad concentrations were about two-orders of magnitude lower than whole-organism concentrations. Consequently, based on our data, assuming whole-organism concentrations

is likely to give a reasonable, or conservative, estimate of amphibian gonad concentrations (as the data of Barnett et al. (2014) demonstrates for mammals).

Although not acknowledged in the accompanying documentation (i.e. Copplestone et al. 2013), the Wildlife Transfer Database does not differentiate between tree components (i.e. data for wood, needles etc. are pooled together to derive

summarised values) and some studies only sample needles (e.g. Thørring et al. 2016) even though arguably the ICRP geometry for their Reference Pine Tree is the trunk wood (ICRP 2008). In Barnett et al. (2014), we suggested that the sampling and analysis of needles would generally result in a conservative $CR_{wo\text{-}soil}$ value. The results from the sites discussed in this paper are supportive of this suggestion, for most elements concentrations in needles and wood (heartwood core) were within an order of magnitude of each other; for I, Na, P and K concentrations in needles were more

than an order of magnitude higher than those in the wood. Only Fe and Ag had concentrations in wood of more than an order of magnitude higher than those in needles; in the case of Fe we have to acknowledge the potential for contamination (from sampling preparation equipment) during the sampling and preparation of the heartwood core samples for analyses.

A potential issue of studies such as that described here is that they generate site-specific data which may not be applicable

to other ecosystems (e.g. Hirth et al. (2017) suggests that $CR_{wo\text{-}soil}$ values for some Australian species differs to those from temperate ecosystems). Novel approaches to estimating the transfer of radionuclides to wildlife using phylogenetic or taxonomic models are being investigated (e.g. Beresford et al. 2013, 2016; Brown et al. 2016; Beresford & Willey 2019). These models would have a number of advantages over the $CR_{wo\text{-}media}$ approach, the most significant being that they account for the influence of site on radionuclide transfer (see Beresford & Willey 2019 for a discussion). The

parameterisation of these models requires data for multiple species from a single site and as such, the data from the study described in this paper are ideal (data presented here for Pb have already been used to establish such a model for Pb transfer to terrestrial wildlife (Beresford & Willey 2019)).

Data presented here will be directly useful in conducting assessments of the exposure of poorly studied organisms. For

example, Tagami et al. (2018) assessed the radiation exposure of frogs in Fukushima Prefecture. Whilst the authors had data for adult and tadpole life-stages, no data were available for frogspawn and consequently the provisional $CR_{wo\text{-}water}$ value from our study was used to estimate activity concentrations in frogspawn based on available measurements of activity concentrations in water.

## 5. Data availability

The data described here(Barnett et al. 2020) have a Digital Object Identifier (https://doi.org/10.5285/8f85c188-a915-46ac-966a-95fcb1491be6) and are freely available from the NERC-Environmental Information Data Centre (https://eidc.ac.uk/) under the Open Government Licence.

## 6. Appendices

**Table A1.** Summary of elements and radionuclides for which data are available by sample type.

| Media or Latin species or family name | Elements and gamma-emitting radionuclides where data are available |
|---|---|
| Soil | Ag, Al, As, Ba, Be, Ca, Cd, Co, Cr, Cs, Cu, Fe, I, K, Li, Mg, Mn, Mo, Na, Ni, P, Pb, Rb, Se, Sr, Ti, Tl, U, V, Zn, $^{40}$K and $^{137}$Cs |
| Water | Al, As, B, Ba, Be, Ca, Cd, Ce, Co, Cr, Cs, Cu, Fe, K, La, Li, Mg, Mn, Mo, Na, Ni,  Pb, Pr, Rb, Sb, Se, Si, Sn, Sr, Ti, U, V, W, Zn, $^{40}$K and $^{137}$Cs |
| *Bombus* spp. | Ag, Al, As, Ba, Be, Ca, Cd, Co, Cr, Cs, Cu, Fe, I, K, Li, Mg, Mn, Mo, Na, Ni, P, Pb, Rb, Se, Sr, Ti, Tl, U, and Zn |
| *Capreolus capreolus* | Ag, Al, As, Ba, Be, Ca, Cd, Co, Cr, Cs, Cu, Fe, I, K, Li, Mg, Mn, Mo, Na, Ni, P, Pb, Rb, Se, Sr, Ti, Tl, U, V, Zn and $^{137}$Cs |
| Lumbricidae | Ag, Al, As, Ba, Be, Ca, Cd, Co, Cr, Cs, Cu, Fe, I, K, Li, Mg, Mn, Mo, Na, Ni, P, Pb, Rb, Se, Sr, Ti, Tl, U, V and Zn |
| *Rana temporaria* | Ag, Al, As, Ba, Be, Ca, Cd, Co, Cr, Cs, Cu, Fe, I, K, Li, Mg, Mn, Mo, Na, Ni, P, Pb, Rb, Se, Sr, Ti, Tl, U, V and Zn |
| *Rana* sp. spawn | Ag, Al, As, Ba, Be, Ca, Cd, Co, Cr, Cs, Cu, Fe, I, K, Li, Mg, Mn, Mo, Na, Ni, P, Pb, Rb, Se, Sr, Ti, Tl, U, V, Zn, $^{40}$K and $^{137}$Cs |
| *Bufo bufo* | Ag, Al, As, Ba, Be, Ca, Cd, Co, Cr, Cs, Cu, Fe, I, K, Li, Mg, Mn, Mo, Na, Ni, P, Pb, Rb, Se, Sr, Ti, Tl, U, V and Zn |
| *Apodemus sylvaticus* | Ag, Al, As, Ba, Ca, Cd, Co, Cr, Cs, Cu, Fe, I, K, Li, Mg, Mn, Mo, Na, Ni, P, Pb, Rb, Se, Sr, Ti, Tl, U, V and Zn |
| *Myodes glareolus* | Ag, Al, As, Ba, Be, Ca, Cd, Co, Cr, Cs, Cu, Fe, I, K, Li, Mg, Mn, Mo, Na, Ni, P, Pb, Rb, Se, Sr, Ti, Tl, U, V and Zn |
| *Abies alba*, *Pinus sylvestris* and *Picea sitchensis* | Ag, Al, As, Ba, Be, Ca, Cd, Co, Cr, Cs, Cu, Fe, I, K, Li, Mg, Mn, Mo, Na, Ni, P, Pb, Rb, Se, Sr, Ti, Tl, U, V,  Zn, $^{40}$K and $^{137}$Cs |
| *Molina caerulea*, *Agrostis giganteana* and *Deschampsia flexuosa* | Ag, Al, As, Ba, Be, Ca, Cd, Co, Cr, Cs, Cu, Fe, I, K, Li, Mg, Mn, Mo, Na, Ni, P, Pb, Rb, Se, Sr, Ti, Tl, U, V,  Zn, $^{40}$K and $^{137}$Cs |
| *Achillia millefolium, Alopecurus myosuroides, Calluna vulgaris, Centaurea nigra, Cirsium palustre, Cirsium vulgare, Digitalis purpurea, Erica tetralix, Juncus effuses, Luzula luzuloides, Polytrichum commune* | Ag, Al, As, Ba, Be, Ca, Cd, Co, Cr, Cs, Cu, Fe, I, K, Li, Mg, Mn, Mo, Na, Ni, P, Pb, Rb, Se, Sr, Ti, Tl, U, V, Zn , $^{40}$K and $^{137}$Cs |
| *Vaccinium myrtillus, Vaccinium* sp., *Calvatia excipuliformis, Clitocybe sp., Collybia sp., Cortinarius sp., Galerina sp., Galerina vittiformis, Gymnopilus sp., Hygrophoropsis aurantiaca, Hygrophorus sp., Laccaria laccata, Mycena epipterygia, Psathyrella sp., Russula emetic, Suillus sp.* | $^{40}$K and $^{137}$Cs |

## 7. Author contribution

CLB arranged sampling sites, led field studies and data compilation, conducted some sample preparation and analysis and made the initial draft of the manuscript. NAB helped define the study protocols, participated in the fieldwork and sample preparation and wrote the final draft of the paper with CLB. MDW helped define the study protocols, participated in the fieldwork and sample preparation and commented on the manuscript. MI conducted the sample analysis and commented on the manuscript. LAW participated in the field sampling and assisted with the initial processing of samples. RF participated in the field sampling, assisted with the initial processing of samples and commented on the manuscript.

## 8. Competing interests

The authors declare that they have no conflict of interest.

## 9. Access and conditions of use

Data are made available under the terms of the Open Government Licence.

## 10. Acknowledgements

This study was funded by the NERC, Environment Agency and Radioactive Waste Management Ltd. under the RATE programme as part of the TREE project (https://tree.ceh.ac.uk/; grant code: NE/L000318/1). The authors are grateful to the Forestry Commission for permission to use the sampling sites and for their assistance in obtaining samples. We would also like to thank various staff members of UKCEH: Claire Wells for assistance during sample preparation and analysis; Claire Carvell for identifying the bee species; Alan Lawlor and Sarah Thacker for ICP-MS analysis of the water samples; and Jacky Chaplow for QC of the accompanying dataset. We also thank Scott Young of University of Nottingham for advice and assistance with sample analysis.

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
