# Peer review of "Element and radionuclide concentrations in soils and wildlife from forests in north-east England with a focus on species representative of the ICRP's Reference Animals and Plants"

_Earth System Science Data, 2020_

## Referee Comment (RC1) · Che Doering (Referee) · 17 Jul 2020

This is a well written and well detailed paper describing a useful and good quality radioecological dataset. There are no major issues with the paper. Some minor technical corrections are suggested below to enhance the paper.

Line 23: The abbreviation CRwo-media is defined at line 15 so does not need to be redefined here.

Line 24: The data will likely contribute to the development of reference databases

under the ICRP's environmental protection framework but not to the development of the framework itself.

Lines 105 and 247: Spelling 'by-catch'.

Line 118: replace the word 'two' with the numeral '2'.

Line 145: Delete the word 'prior'.

Lines 410-411: If available, a reference should be provided to an IAEA proficiency test report to substantiate the claim being made in this sentence.

Line 517: Grammar 'analyse'.

Lines 529 and 530: replace 'FM' with 'filtered'.

---

## Author Comment (AC1) · 24 Jul 2020

We thank the reviewer for their positive comments. Their suggested corrections will be made in the final manuscript once we receive comments from other reviewers.
* * *

---

## Referee Comment (RC2) · Anonymous Referee #2 · 11 Aug 2020

This is a very well written paper and the comments are to enhance clarity and to provide further explanation of some of the statements made. Line 50: it would be helpful to very briefly explain the 'Reference Site' concept. Line 101 onwards. Somewhere there needs to be a list of the elements/nuclides for which measurements have been made - this is only mentioned in Section 2.4. It would also be helpful to include a table that indicates which elements/nuclides are measured for each species. Line 215 and 253. Please explain why samples for these samples were ashed and others were freeze-dried in the preparation methodology. Line 400 (and earlier). What was the

justification for choice of species for gamma analysis? Line 404/405. It is not clear from this sentence that only frogspawn was analysed for gammas and not other frog flesh samples. Line 531-Section 4. Please can the authors comment on the limitations of using location specific soil concentrations for estimating CR values for animals that graze a large area eg deer. How is this addressed in this and in general. Line 558 onwards. As the authors suggest, it would not be expected that the CR values for Cs-137 would be up to an order of magnitude higher than stable Cs and this deserves a note that further investigation is warranted for potentially important nuclides with respect to dose. Other published reports eg IAEA TRS479 make a comment that using CR for stable elements is likely to be conservative for short-lived nuclides if equilibrium is not reached and could be cited. Line 581. Sentence is not clear and would benefit from splitting.

---

## Author Comment (AC2) · 12 Aug 2020

We thank the reviewer for their positive comments. Their suggested corrections will be made in the final manuscript once we have been notified discussion has closed.

---

## Referee Comment (RC3) · Anonymous Referee #3 · 15 Aug 2020

Review of Element and radionuclide concentrations in soils and wildlife from forests in north-east England with a focus on species representative of the ICRP's Reference Animals and Plants

Overall, this work provides an important set of data for the ICRP Raps. This effort is in line with what is needed to support and improve on the ICRP framework for assessing radiological impacts to the environment.

This reviewer has no major issues with the manuscript. Suggested below are several

areas where additional information is needed. This paper may be referenced often, and may serve as a guide to others (potentially students or new researchers to the field in the future) and I believe the points raised below should be clarified.

Comments:

Line 48, ICRP's Line 60. The text above says that dose assessment is the underlying motive. Should mention here (or in discussion later) whether the radionuclides/elements chosen for this study are in fact important for dose assessment. Lin 60. If radionuclides of importance to biota dose were excluded (e.g. Ra226), it should be noted and discussed. Some readers may consider this study as a pattern for future studies and they would benefit from a statement about the need to prioritise data gaps on the radionuclides important for dose. Line 128. Need a bit more on soil sampling methods. The word "approximately" is used when describing the "15 cm x 15 cm x 10 cm deep" samples.

–Need to state that the soil samples were gathered using a method that ensured all depths 0-10cm were equally represented.

–If a coring device/tube wasn't used, the authors should acknowledge and estimate the potential uncertainty on CRs that result. For example, if the soils were gathered using a shovel/spade it is likely the surface layers were over-represented (by mass) which could influence study results for Cs-137 and other anthropogenics. Line 129. "locations corresponded to the sampling sites of the animals and plants collected." Describe distance between plants and soil samples and if they were 1:1 soil-plant pairs or 3:1 pairs, etc.

Line 129. How organic vs mineral soils are sampled may influence the calculated CRs. Describe if organic material was removed from the ground surface before soil sampling, and if so, how much was removed. Fallout radionuclides accumulate in detritus and organic soils (numerous references). How did you determine where the organic detritus ends and soil begins?

Line 135. "Once dry a sub-sample was manually homogenised and" This seems backwards. The entire sample should have been homogenised before a subsample was removed.

Line 144. "Three water samples (each of approximately 1 litre) were collected from the Kielder main sampling site on 16 March 2015. The samples were collected from three areas within the. . ." Describe the water body. River, pond, lake?

Line 183. ". . .paper to allow for gut evacuation." State the length of the depuration period.

Line 269. ". . .both leaf and flower stems were collected." State how close to the roots/ground the grass samples were cut.

Line 563. "This perhaps raises a question with regard to using stable element CRwo-soil values, especially when they are used to represent shorter-lived radioisotopes, within radiological assessments that typically aim to be conservative." Thank you for the discussion on stable Cs vs radiocaesium. Your statement here can/should be more definitive. Your data does more than raise the question (not perhaps).

–Suggest you state that your data demonstrate that the elemental Cs and anthropogenic Cs-137 uptake are not equal as has been assumed in some past studies.

–Suggest you state that your data indicate use of stable element data may introduce further CR uncertainty when applied to radionuclides.

–Your discussion explaining the above focuses on half-lives, when (as I know the authors know) it should focus on physico-chemical differences between the stable elements vs anthropogenics.

Line 567. "4.1 Use of data" Somewhere in this discussion on use of data, it would be fair to point out that these CRs of this study are specific to/representative of a temperate forest ecosystem and that it has been indicated that CRs developed for other ecosystems may vary (e.g. higher CRs for arid system were indicated in Hirth et
al. 2019, some differences were reported for uptake in Japan vs Europe in Tagami et al. 2017).

---

## Author Comment (AC3) · 25 Aug 2020

We thank the reviewer for their comments. We will provide further clarification on the points they have raised in the final manuscript once we have been notified discussion has closed.

---

## Author Response (AR1)

**Point by point response to reviewers**

**Element and radionuclide concentrations in soils and wildlife from forests in north-east England with a focus on species representative of the ICRP's Reference Animals and Plants**

We would like to thank the three reviewers for their positive comments on our paper. Below we detail our response to each of the points they raise.

*Reviewer 1*

Line 23: The abbreviation $CR_{wo\text{-}media}$ is defined at line 15 so does not need to be re-defined here.

RESPONSE>>Deleted as suggested.

Line 24: The data will likely contribute to the development of reference databases under the ICRP's environmental protection framework but not to the development of the framework itself.

RESPONSE>>Text re-phrased within the abstract to reflect this comment.

Lines 105 and 247: Spelling 'by-catch'.

RESPONSE>>Corrected in sections 2.2.7 and 2.2.9.

Line 118: replace the word 'two' with the numeral '2'.

RESPONSE>>Amended as suggested in section 2.2.

Line 145: Delete the word 'prior'.

RESPONSE>>Deleted as suggested.

Lines 410-411: If available, a reference should be provided to an IAEA proficiency test report to substantiate the claim being made in this sentence.

RESPONSE>>Links to the IAEA and other proficiency schemes are now included within section 2.5.

Line 517: Grammar 'analyse'.

RESPONSE>>Corrected.

Lines 529 and 530: replace 'FM' with 'filtered'

RESPONSE>>Corrected within Section 3.

*Reviewer 2*

Line 50: it would be helpful to very briefly explain the 'Reference Site' concept.

RESPONSE>>Text amended to better reflect wording in ICRP Publication 114 and subsequent usage within Section 1.

Line 101 onwards. Somewhere there needs to be a list of the elements/nuclides for which measurements have been made - this is only mentioned in Section 2.4.

RESPONSE>>List added for stable elements towards the end of the Introduction (where radionuclides were already listed).

It would also be helpful to include a table that indicates which elements/nuclides are measured for each species.

RESPONSE>>A summary table has been added as Appendix 1.

Line 215 and 253. Please explain why samples for these samples were ashed and others were freeze-dried in the preparation methodology.

RESPONSE>>Explanation added to sub-section 2.2.7.

Line 400 (and earlier). What was the justification for choice of species for gamma analysis?

RESPONSE>>Explanation (available sample size) added to section 2.5.

Line 404/405. It is not clear from this sentence that only frogspawn was analysed for gammas and not other frog flesh samples.

RESPONSE>>Frog flesh tissues were of insufficient size (hopefully addressed within edit made in response to the previous question regarding which elements/nuclides are measured for each species). See Appendix 1 and Section 2.5.

Line 531-Section 4. Please can the authors comment on the limitations of using location specific soil concentrations for estimating CR values for animals that graze a large area e.g. deer. How is this addressed in this and in general.

RESPONSE>>To best address this comment we have added text to Section 2.2.1 re sampling soils from within a typical home range area for roe deer (a relatively small species of deer which typically have relatively small home ranges (0.5 km$^2$ for females in the UK).

Line 558 onwards. As the authors suggest, it would not be expected that the CR values for Cs-137 would be up to an order of magnitude higher than stable Cs and this deserves a note that further investigation is warranted for potentially important nuclides with respect to dose. Other published reports eg IAEA TRS479 make a comment that using CR for stable elements is likely to be conservative for short-lived nuclides if equilibrium is not reached and could be cited.

RESPONSE>>See Section 4. In considering the reviewers comments we have deleted '*especially when they are used to represent shorter-lived radioisotopes*' from the manuscript as the comment that our results raise questions with respect to the application of stable element data to radiological assessment is generic (Cs-137 and Sr-90 as discussed in our paper are not 'short-lived' radionuclides).

The comment in IAEA TRS479 which the reviewer refers to is actually a different point – it eludes to the fact that short-lived radionuclides (e.g. those with a physical half-life of days) decay away before soil to organism transfer can reach equilibrium. Therefore an equilibrium CR (based on stable data or indeed, if available, that from longer-lived radioisotopes of the same element) would be conservative. NOTE a number of the authors of this manuscript were leading authors/contributors to IAEA TRS479.

Line 581. Sentence is not clear and would benefit from splitting.

RESPONSE>>Sentence has been split as suggested. See Section 4.1.

*Reviewer 3*

Line 48, ICRP's Line 60. The text above says that dose assessment is the underlying motive. Should mention here (or in discussion later) whether the radionuclides/elements chosen for this study are in fact important for dose assessment.

RESPONSE>>A note that the majority of elements have radioisotopes which need to be considered in radiological assessments has been added (following list of elements reported) within the Introduction.

Line 60. If radionuclides of importance to biota dose were excluded (e.g. Ra226), it should be noted and discussed. Some readers may consider this study as a pattern for future studies and they would benefit from a statement about the need to prioritise data gaps on the radionuclides important for dose.

RESPONSE>>The reviewer makes a valid point and we have added a paragraph to Section 4 to discuss this.

Line 128. Need a bit more on soil sampling methods. The word "approximately" is used when describing the "15 cm x 15 cm x 10 cm deep" samples.

–Need to state that the soil samples were gathered using a method that ensured all depths 0-10cm were equally represented.

–If a coring device/tube wasn't used, the authors should acknowledge and estimate the potential uncertainty on CRs that result. For example, if the soils were gathered using a shovel/spade it is likely the surface layers were over-represented (by mass) which could influence study results for Cs-137 and other anthropogenics.

RESPONSE>>We have expanded on the sampling protocol within Section 2.2.1 to state that surface layers were not over-represented by mass (or indeed volume).

Line 129. "locations corresponded to the sampling sites of the animals and plants collected." Describe distance between plants and soil samples and if they were 1:1 soil-plant pairs or 3:1 pairs, etc.

RESPONSE>> We have added  text to section 2.7 on averaging of soil concentrations and for every CR value in the accompanying dataset (https://doi.org/10.5285/8f85c188-a915-46ac-966a-95fcb1491be6) the soil concentrations used are clearly identified (columns labelled e.g. 'Co_Located_Soil_Sample_Used_To_Calculate_Concentration_Ratio'.

Line 129. How organic vs mineral soils are sampled may influence the calculated CRs. Describe if organic material was removed from the ground surface before soil sampling, and if so, how much was removed. Fallout radionuclides accumulate in detritus and organic soils (numerous references). How did you determine where the organic detritus ends and soil begins?

RESPONSE>> Information added to section 2.2.1.

Line 135. "Once dry a sub-sample was manually homogenised and" This seems backwards. The entire sample should have been homogenised before a subsample was removed.

RESPONSE>>Paragraph has been reworded to better reflect sample processing.

Line 144. "Three water samples (each of approximately 1 litre) were collected from the Kielder main sampling site on 16 March 2015. The samples were collected from three areas within the. . ." Describe the water body. River, pond, lake?

RESPONSE>> Collection was from 'seasonal ponds' - information added to section 2.2.2.

Line 183. ". . .paper to allow for gut evacuation." State the length of the depuration period.

RESPONSE>> '48 hours' – information added to section 2.2.5.

Line 269. ". . .both leaf and flower stems were collected." State how close to the roots/ground the grass samples were cut.

RESPONSE>> 'to within c. 1 cm of the soil surface' – information added to section 2.2.11.

Line 563. "This perhaps raises a question with regard to using stable element $CR_{wosoil}$ values, especially when they are used to represent shorter-lived radioisotopes, within radiological assessments that typically aim to be conservative." Thank you for the discussion on stable Cs vs radiocaesium. Your statement here can/should be more definitive. Your data does more than raise the question (not perhaps).

–Suggest you state that your data demonstrate that the elemental Cs and anthropogenic Cs-137 uptake are not equal as has been assumed in some past studies.

–Suggest you state that your data indicate use of stable element data may introduce further CR uncertainty when applied to radionuclides.

–Your discussion explaining the above focuses on half-lives, when (as I know the authors know) it should focus on physico-chemical differences between the stable elements vs anthropogenics.

RESPONSE>> Text has been amended to be more definitive, also taking into account a comment from reviewer #2.

Line 567. "4.1 Use of data" Somewhere in this discussion on use of data, it would be fair to point out that these CRs of this study are specific to/representative of a temperate forest ecosystem and that it has been indicated that CRs developed for other ecosystems may vary (e.g. higher CRs for arid system were indicated in Hirth et al. 2019, some differences were reported for uptake in Japan vs Europe in Tagami et al. 2017).

RESPONSE>> We amended the text to acknowledge site specific nature of $CR_{wo}$ values using Hirth et al. (2017) as an example. We have also emphasised that the values we present were collected from temperate forest ecosystems.

AUTHOR>Minor grammatical errors have also been corrected and re-ordering of the reference list to alphabetical.

[revised manuscript text omitted]